# Human umbilical cord mesenchymal stem cells decellular matrix alleviates chondrocyte senescence by inhibiting the STING-NF-κB pathway

Yuelong Zhang⊕, Xunshan Ren⊕, Huangming Zhuang⊕, Huajie Li, Miradj Siddick Adam, Rongling Feng, Junming Zhu, Panghu Zhou ⓘ *

Department of Orthopedics, Renmin Hospital of Wuhan University, Wuhan, China

⊕ These authors contributed equally to this work.
* zhoupanghu@whu.edu.cn

## Abstract

Osteoarthritis (OA), characterized by synovial inflammation, articular cartilage degeneration, and structural changes of subchondral bone and periarticular tissues, represents a major unmet clinical challenge. Targeting senescent chondrocytes has emerged as a promising therapeutic strategy of OA. Human umbilgratical cord mesenchymal stem cells (hUCMSCs) have shown potential in OA treatment through paracrine mechanisms, but their clinical translation is limited by challenges in cell viability control and safety concerns. hUCMSCs decellular extracellular matrix (hUCMSCs-dECM) can target senescent chondrocytes to alleviate senescence in OA. Stimulator of interferon gene (STING) can promote chondrocyte senescence in OA through the activation of NF-κB signaling, and inhibition of STING may provide a novel approach for OA treatment. Here, we demonstrated that hUCMSCs-dECM attenuated chondrocyte senescence *in vivo* and *in vitro* by inhibiting the STING-NF-κB pathway, which would provide a novel approach for OA treatment.

## 1. Introduction

Osteoarthritis (OA), a prevalent degenerative joint disorder, is characterized by synovial inflammation, articular cartilage degeneration, and structural alterations in the subchondral bone, ligaments, and periarticular muscles [1]. Driven by multifactorial pathogenesis, end-stage OA culminates in global joint destruction and functional impairment, imposing substantial socioeconomic burdens on healthcare systems [2]. During OA progression, the articular cartilage matrix undergoes marked structural degradation, manifesting as surface fibrosis, fragment detachment, and deep fissure formation [3]. Notably, chondrocytes in OA exhibit a senescence-associated secretory phenotype (SASP), characterized by sustained production of reactive oxygen species (ROS), cytokines, and other pro-inflammatory mediators [4]. This phenomenon

**Data availability statement:** All relevant data are within the manuscript and its Supporting Information files.

**Funding:** This research was funded by Panghu Zhou of the National Natural Science Foundation of China (grant number. 82372489), the Wuhan University Education and Development Foundation (grant number. 2002330), the Fundamental Research Funds for the Central Universities (grant number. 2042023kf0224), and the Cross-Innovation Talent Program of Renmin Hospital of Wuhan University (grant number. JCRCFZ-2022-019). The funders had no role in study design, data collection and analysis, decision to publish, or preparation of the manuscript.

**Competing interests:** The authors have declared that no competing interests exist.

suggests that chondrocyte senescence may play a pivotal regulatory role in OA pathogenesis, raising a critical scientific question of whether it is possible to alleviate OA by delaying chondrocyte senescence and to develop new therapeutic approaches to OA.

Cellular senescence entails irreversible cell cycle arrest accompanied by bioactive SASP secretion, comprising inflammatory cytokines, growth factors, proteases and chemokines [5–7]. Over the past many years, researchers have begun to explore the treatment of OA by targeting chondrocytes and other joint tissue cells undergoing cellular senescence [8]. Jeon et al. demonstrated that senescent chondrocytes inhibit cartilage formation in neighboring non-senescent cells via extracellular vesicle (EV)-mediated paracrine signaling. Pharmacological clearance of senescent chondrocytes significantly attenuated EV production, thereby ameliorating OA pathology [9]. Complementary studies by Ji et al. revealed that pharmacological intervention in chondrocyte senescence effectively blocks OA progression [10,11]. These findings collectively demonstrate the therapeutic potential of targeting senescent chondrocytes for OA prevention and treatment.

MSCs have been extensively studied as a therapeutic platform since their first clinical trials in 1995 and are now among the most clinically investigated cell therapies worldwide [12]. hUCMSCs have gained prominence due to their high proliferative capacity, ease of accessibility, and low immunogenicity [13]. Both preclinical and clinical studies have demonstrated the therapeutic potential of hUCMSCs in OA, with evidence suggesting that their secreted cytokines promote endogenous tissue repair [14–16]. Beyond cell-based therapies, innovative strategies such as combining hUCMSCs with biologics or administering hUCMSC-derived EV have shown significant efficacy in OA treatment [17–20].

The extracellular matrix (ECM) derived from hUCMSCs (hUCMSCs-dECM) is rich in collagen I and glycosaminoglycans (GAGs), key components critical for restoring tissue homeostasis post-injury [21,22]. Recent studies highlight the broad therapeutic potential of dECM in senescence-associated disorders, including myocardial injury, acute kidney injury, and intervertebral disc degeneration [23–26]. Notably, hUCMSC-derived ECM has emerged as a promising scaffold for regenerative applications. For instance, Liu et al. [27] demonstrated that hUCMSCs-dECM enhances Schwann cell proliferation and neurite outgrowth, offering a neural niche-mimicking platform for peripheral nerve regeneration. Similarly, Zhang et al. [28] reported that 3D hUCMSC-dECM substrates markedly improve chondrocyte proliferation and differentiation compared to traditional 2D cultures, preserving chondrogenic phenotype in vitro. Complementing these findings, Khan et al. [29]developed decellularized umbilical cord hydrogels that robustly support MSC chondrogenesis, upregulating key markers like SOX-9 and aggrecan. Despite these advances, the mechanistic role of hUCMSCs-dECM in targeting cellular senescence—particularly in OA—remains underexplored. While alternative biomaterials, such as biotechnological chondroitin (BC) and hydrogels, have shown promise in OA treatment by enhancing collagen II synthesis and mitigating inflammation [30,31], hUCMSCs-dECM offers unique advantages. Unlike extractive chondroitin sulfate (CS), BC derived from engineered

processes demonstrates superior anti-inflammatory effects and phenotype preservation in chondrocytes [30]. Hydrogels, though versatile, often lack the tissue-specific ECM composition necessary to fully replicate the native cartilage micro-environment [31]. In contrast, hUCMSCs-dECM inherently retains chondroinductive cues, positioning it as a scaffold that bridges the gap between biological functionality and structural support. However, critical questions persist: How does hUCMSCs-dECM mechanistically alleviate chondrocyte senescence in OA? This makes one wonder how the therapeutic effect and mechanism of dECM are in treating cellular senescence-related diseases and whether hUCMSCs-dECM can target senescent chondrocytes to alleviate senescence in OA.

The stimulator of interferon genes (STING) a sensor on bacterial and viral cell membranes, initiates downstream reactions via the cyclic GMP-AMP synthase (cGAS)-STING pathway upon DNA damage by recognizing cytoplasmic DNA [32–34]. Recent studies demonstrat that STING activation triggers downstream NF-κB signaling [35,36]. Following STING activation, TBK1 and its homolog IκB kinase epsilon (IKKε) activate the IKK complex, subsequently inducing NF-κB transcription and triggering a pro-inflammatory cytokine response [37,38]. Pan et al. reported that inhibiting the STING-NF-κB pathway reduced inflammation and psoriasis severity in keratinocytes and immune cells [39]. Sun et al. showed that Nrf2 activators suppress NF-κB signaling by blocking TRAF6 recruitment to STING, mitigating osteoclastogenesis and ovariectomy-induced bone loss in vivo [40]. Guo et al. further demonstrated that STING activates NF-κB signaling cascades *in vitro*, while NF-κB blockade attenuated STING-induced apoptosis, senescence, and metabolic imbalance. *In vivo*, STING knockdown alleviated medial meniscus instability-driven OA development in mice [32]. Collectively, these findings indicate that STING promotes OA via NF-κB activation, and its inhibition offers a novel therapeutic avenue.

In this study, we demonstrated that hUCMSCs-dECM attenuates chondrocyte senescence and SASP *in vitro* and *in vivo* by inhibiting the STING-NF-κB pathway, presenting a novel therapeutic strategy for OA.

## 2. Materials and methods

### 2.1. Isolation and culture of hUCMSCs

Purchased hUCMSCs were cultured in MEMα medium supplemented with 10% fetal bovine serum (FBS, Hyclone), 100 U/mL penicillin, and 100 U/mL streptomycin (Servicebio) and incubated at 37°C in a humidified atmosphere of 5% $CO_2$. The medium was changed every 3 days. For cytometric characterization, cells at passage 3 were harvested and stained with the following antibodies against CD34/44/45/73/90/105. Flow cytometry was performed using a CytoFlex flow cytometer, and data were analyzed with FlowJo v10.8.1. Cells showing >95% positivity for CD44/73/CD90/CD105 and <2% for CD34/CD45 were used for experiments. Cultured cells were tested for the characterization of surface markers at the 3rd passaging and cultured with osteogenic, chondrogenic, and lipogenic differentiation media to verify their trilineage differentiation ability. Primary hUCMSCs were cultured for 4–6 generations for the next experiment.

### 2.2. hUCMSCs-dECM solution preparation

hUCMSCs at passages 4–6 were seeded at a density of $1 \times 10^6$ cells/cm² in 150 mm culture dishes and cultured for 7 days in MEMα medium supplemented with 50 µg/mL ascorbic acid (Sigma) to promote ECM deposition. After decellularization with 3% Triton X-100 (Biosharp) solution for 48 h, the remaining cellular debris was removed by washing with distilled water (3 × 30 min), followed by DNase/RNase treatment (Servicebio, 37°C, 2 h) to eliminate residual nucleic acids. The decellularized ECM was lyophilized, and the yield was quantified as $0.5 \pm 0.1$ mg/cm² (n = 5 batches). After centrifugation, the dECM was frozen at −80°C. To make the hUCMSCs-dECM solution, lyophilized dECM was dissolved in distilled water to configure different concentration gradient solutions for subsequent experiments.

### 2.3. Scanning electron microscopy (SEM)

SEM images of dECM hydrogels were taken to examine their microstructural morphology. The samples were fixed using 2.5% glutaraldehyde (Sigma) and washed with phosphate-buffered saline. The washed samples were then rinsed three

times in deionized water to remove residual impurities. The samples were dehydrated in a series of increasing ethanol gradients until they were treated with 100% ethanol. Samples were observed using a scanning electron microscope (Sigma) and images were analyzed using Smart SEM software.

## 2.4. Cytotoxicity assessment

To measure the cytotoxicity of hUCMSCs-dECM, we inoculated 5000 chondrocytes per well in a 96-well plate. After adherence, the medium was replaced with fresh medium containing different concentrations of hUCMSCs-dECM (0, 1, 2, 5, 10, 20, 50, 100, 200 μg/mL) and incubated at 37°C for 24 h, 48 h. Next, 10 μL of Cell Counting Kit-8 (Servicebio) reagent was added to each well and incubated at incubated for 2 h at 37°C. Absorbance at 450 nm was measured using an enzyme marker (EnVision, USA).

## 2.5. Isolation and culture of chondrocytes

Our animal experiments were performed by protocols approved by the Hubei Provincial Animal Care and Use Committee (Approval No: 20230101A). We isolated knee cartilage tissue from Wistar rats (8 weeks, male) to isolate primary chondrocytes. Cartilage was minced and digested with 0.2% type II collagenase (Servicebio) overnight in a 37°C incubator with 5% $CO_2$. Chondrocytes were collected by centrifugation at 1200 RPM for 5 min and resuspended in DMEM/F12 (Servicebio) containing 10% fetal bovine serum (FBS, Gibco), 1% penicillin and streptomycin (Servicebio). Second or third-generation chondrocytes were used in our study.

## 2.6. Real-time quantitative polymerase chain reaction (RT-qPCR)

Chondrocytes were inoculated in 6-well plates and incubated for 24 hours. After stimulation with $H_2O_2$ (500 μM/ml) (Aladdin) for 4 h, the treatment group was supplemented with 10ug/ml hUCMSCs-dECM. total RNA was isolated from chondrocytes using an RNA isolation kit with a centrifugal column (Beyotime). Total RNA was reverse transcribed into a 20 μl reaction volume of complementary DNA (cDNA) using a double-stranded cDNA synthesis kit (Servicebio). RT-qPCR amplification was performed in LightCycler® 480 Software Version 1.5 (Roche) using the 2×Universal SYBR green fast qPCR mix kit (Abclonal). Data were analyzed using the 2-ΔΔCT method. Primer sequences for each gene used in this study are listed in S1 Table.

## 2.7. Western blot

Total protein was isolated from chondrocytes cultured using RIPA lysis buffer containing 1 mM PMSF. The samples were incubated on ice for 30 minutes and centrifuged at 12,000 rpm for 10 minutes at 4°C. The supernatant after centrifugation was collected and the protein concentration was determined by BCA protein assay (Servicebio, G2026). Protein samples were then mixed with SDS and incubated at 99°C for 10 min. 50 μg of protein from each sample was loaded onto SDS polyacrylamide gel electrophoresis (SDS-PAGE) and separated by electrophoresis. Proteins were transferred to PVDF membranes incubated with primary antibodies against IL-6 (1:1000, Abmart), iNOS (1:1000, Abclonal), COX-2 (1:1000, Abacm), MMP3 (1:1000, Abclonal), MMP13 (1:1000, Abclonal), ADAMTS5 (1:1000, Abclonal), IKKβ (1:1000 CellSignalingTechnology), p-IKKβ (1:1000, CellSignalingTechnology), IκBα (1:1000, Abmart), p-IκBα (1:1000, Abmart), p65 (1:1000, Abmart), p-p65 (1:1000, Abmart), and β-actin (1:3000, Servicebio) were incubated at 4°C overnight. Next, the membrane was incubated with enzyme-linked secondary antibody (1:3000, Servicebio) for 1 hour at room temperature. To visualize the immunoblot, we added enhanced chemiluminescence (ECL, Epizyme). All bands were photographed using a Bio-Rad scanner and quantified by Image-J software (version 1.8).

## 2.8. Immunofluorescence staining

Second or third-generation chondrocytes were inoculated and incubated on glass coverslips for 24 hours. After fixation in 4% paraformaldehyde for 30 min in a room temperature environment, the chondrocytes were washed three times

with PBST (0.1% TritonX-100 in PBS). After that, chondrocytes were closed with 1% bovine serum albumin in PBS for 1 hour. Chondrocytes were washed 3 times with PBST (PBS containing 0.05% Twen-20) and incubated with MMP3 (1:400, Servicebio), MMP13 (1:200, Servicebio), IL-1β (1:500, Servicebio), and IL-6 (1:200, Servicebio), and other antibodies were incubated at 4°C overnight. The samples were then incubated with fluorescein-coupled goat anti-rabbit IgG antibody (1:400) at room temperature for 1 h. DAPI was used to stain the nuclei. Chondrocytes were observed and photographed using a laser confocal microscope. Cell fluorescence intensity was quantified by assessing positive cells using Image-J software (version 1.8).

## 2.9. STING gene silencing

Small interfering RNA (siRNA) targeting human STING (si-STING) and scrambled control (si-NC) were synthesized by GenePharma. Chondrocytes at 60–70% confluency were transfected with 50 nM siRNA using Lipofectamine 3000 (Invitrogen) according to the manufacturer's protocol. Knockdown efficiency was validated 48 h post-transfection by RT-qPCR and Western blots. Cells with >70% STING protein reduction were used for dECM preparation.

## 2.10. OA rat models

All animal procedures were approved by the Ethics Committee of Renmin hospital of Wuhan university (Approval No: 20230101A, Date: January 2023). and conducted in accordance with the National Institutes of Health Guide for the Care and Use of Laboratory Animals.

Fifteen 8-week-old male Wistar rats (weight: 200–220 g) were randomly divided into three groups: sham, OA (saline), and OA+dECM. OA was induced by anterior cruciate ligament transection (ACLT). Briefly, rats were anesthetized with intraperitoneal pentobarbital (40 mg/kg), and the knee joint was exposed through a medial parapatellar incision. The ACL was transected in the OA and dECM groups, while the sham group underwent joint cavity exposure without ligament transection. Postoperatively, buprenorphine (0.05 mg/kg) was administered subcutaneously every 12 hours for 48 hours to alleviate pain and penicillin (40,000 U/kg) was administered intramuscularly daily for 3 days to prevent infection. Rats in the OA+dECM group received weekly intra-articular injections of dECM (10 µg/mL, 100 µL), while the OA group received saline (100 µL). Any animal exhibiting severe distress (>20% weight loss, inability to reach food/water, or self-mutilation) would have been euthanized immediately; no cases required early termination. Eight weeks post-surgery, rats were euthanized by cervical dislocation under deep anesthesia induced with 5% isoflurane in oxygen (flow rate: 2 L/min) to ensure a painless procedure. Knee samples were fixed with 4% paraformaldehyde, decalcified in 0.5 M EDTA, and embedded in paraffin. A portion of the cartilage tissue was used for frozen sections.

## 2.11. Histology

Paraffin-embedded tissues were cut into 6-µm thick sections using a rotary slicer. Sections were then stained in the midsagittal plane with safraninO-fast green stain. Microscopic images were taken using an inverted microscope ((NIKON ECLIPSE CI, NIKON). OA severity was assessed using the Osteoarthritis Research Society International (OARSI) system.

## 2.12. Immunohistochemistry

Sections were subjected to antigen repair in citrate buffer (PowerforBiologu). Non-specific protein binding was blocked using BSA (Solarbio) in PBST for 30 minutes. Next, sections were incubated with anti-P16 (Wanleibio), P21 (Wanleibio), iNOS (Servicebio), COX2 (Abclonal), MMP3 (Abclonal), IL-6 (Abclonal), p65 (Abmart) and p-p65 (Abmart) primary antibodies were incubated at 4°C overnight. After washing with PBS, the sections were incubated with goat anti-rabbit IgGHRP (Abcam) for 50 min at 37°C. A DAB HRP substrate kit (DAKO) was used for dye development, and A DAB HRP substrate kit (DAKO) was used for dye development, and hematoxylin was used as a nuclear counterstain. Microscope

images were acquired with an inverted microscope (NIKON ECLIPSE CI, NIKON). Image Pro Plus (version: 6.0) was used to quantify the results by measuring the mean integral optical density.

## 2.13. Statistics

All experiments in this study were repeated at least three times and data were expressed as mean±standard error. Statistical significance was determined using one-way analysis of variance (ANOVA) (more than two groups) or Student's t-test (two groups). All reported p-values were calculated using two-sided comparisons. $p < 0.05$ was considered statistically significant.

## 3. Results

### 3.1. hUCMSCs-dECM extraction and characterization

We employed a series of streamlined chemical decellularization protocols to extract hUCMSCs from human umbilical cord-derived Wharton's jelly connective tissue, optimizing the extraction time to obtain fresh hUCMSCs-dECM. hUCMSCs were usually cultured for 7 days to facilitate complete dECM production, followed by decellularization with Triton X-100, thorough washing to remove cellular debris, and Dnase/Rnase treatment to eliminate residual DNA/RNA. The resulting dECM was centrifuged and stored at −80°C (Fig 1a).

Flow cytometry was used to characterize the cell surface markers of hUCMSCs, and there was a positive expression of CD44, CD73, CD90, and CD105, and negative expression of CD34 and CD45 (Fig 1b). Adipogenic, osteogenic, and chondrogenic differentiation was validated via Oil Red O, Alizarin Red, and Alcian Blue staining, respectively, confirming the trilineage potential of isolated hUCMSCs (Fig 1c-e). The above experimental evidence showed that the cells had the main characteristics of MSCs, confirming that hUCMSCs had been successfully isolated.

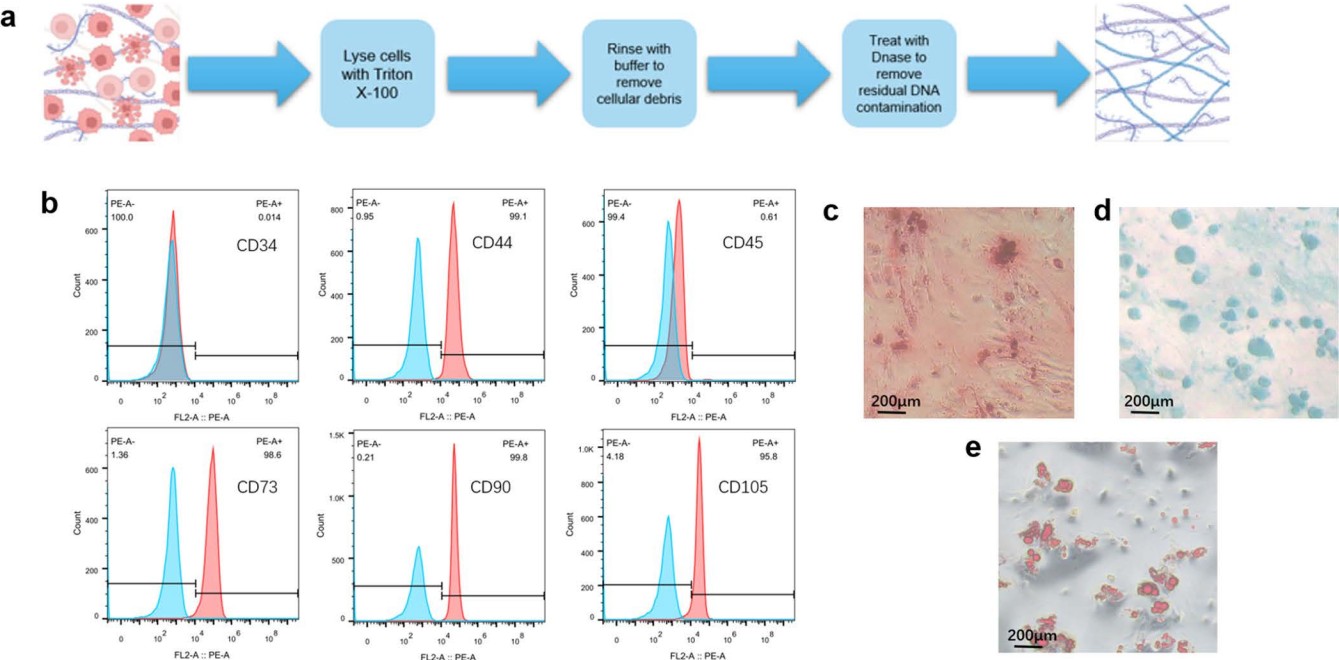

**Fig 1. Extraction process and characterization of hUCMSCs-dECM. (a)** A series of chemical decellularization protocols were taken to extract dECM from hUCMSCs. **(b)** Flow cytometry identification of hUCMSCs. **(c)** Mineralized nodules were stained by Alizarin Red after osteogenic differentiation induction of hUCMSCs. **(d)** Acid mucopolysaccharide was stained by Alcian Blue after chondrogenic differentiation induction of hUCMSCs. **(e)** Lipid droplets were stained by oil red O after adipogenic differentiation induction of hUCMSCs.

## 3.2. hUCMSCs-dECM on chondrocyte viability

Clinical-grade hUCMSCs were obtained from Wingor Biotechnology Co., Ltd. (Shenzhen, China). SEM revealed the morphological integrity of decellularized hUCMSCs-dECM, displaying uniform particle morphology (Fig 2a). CCK-8 assays quantified dECM cytotoxicity, demonstrating no significant reduction in chondrocyte viability at concentrations up to 200 µg/mL over 72 hours (Fig 2b-d). Based on prior studies [23,41], a 10 µg/mL dECM concentration was selected for subsequent experiments..

## 3.3. hUCMSCs-dECM attenuates $H_2O_2$-induced senescence and inhibits chondrocyte SASP

hUCMSCs have demonstrated therapeutic potential in cellular senescence-related diseases, and dECM have shown protective effects against tissue-damage-associated pathologies [42]. To evaluate the anti-senescence efficacy of hUCMSCs-dECM, we established an $H_2O_2$-induced chondrocyte senescence model and compared its effects with a commercial ECM (Corning® Matrigel®, Cat# 356234).

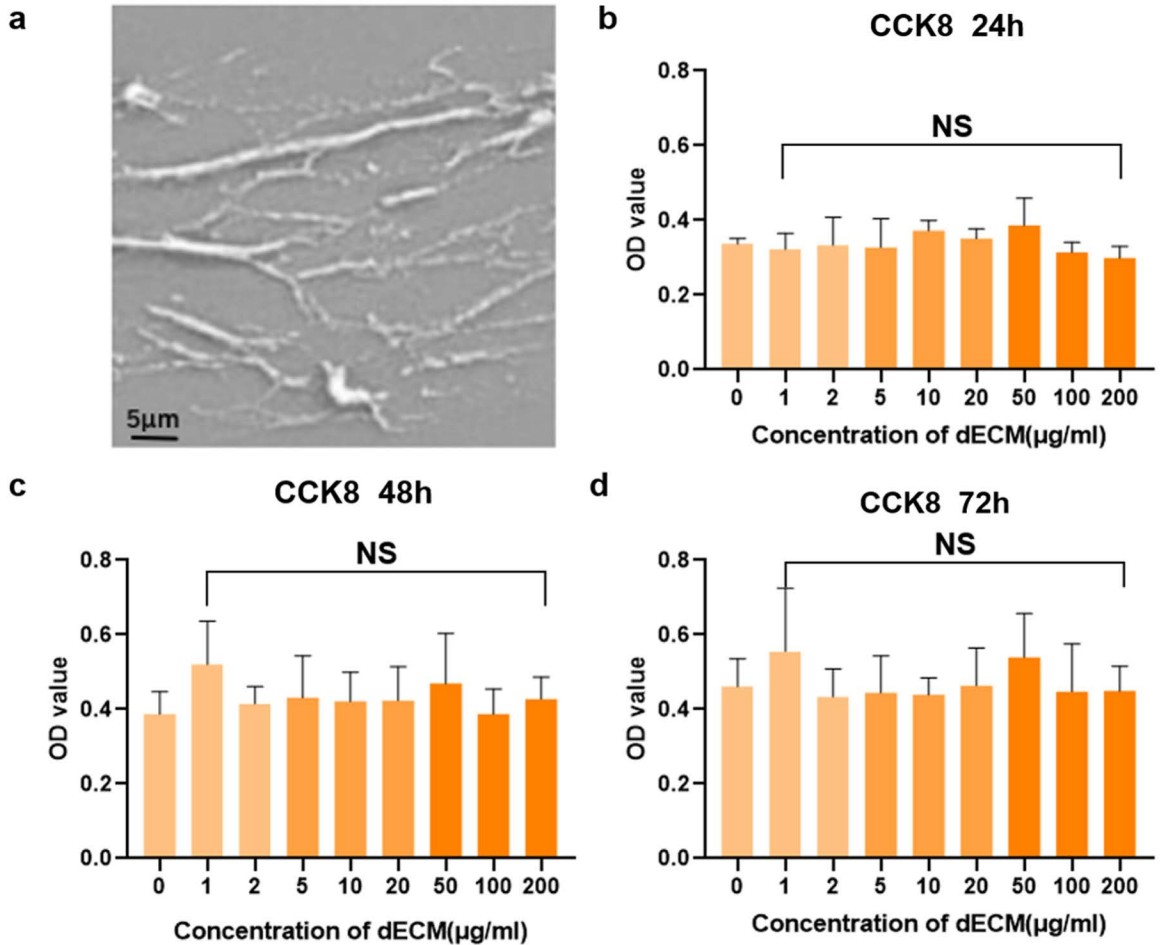

**Fig 2. Effect of hUCMSCs-dECM on cell viability. (a)** SEM images of hUCMSCs-dECM. **(b-d)** Cytotoxicity of hUCMSCs-dECM on chondrocytes was determined using the CCK-8 assay and incubated at different concentrations for 24 h, 48 h, and 72h. The cytotoxicity of hUCMSCs-dECM on chondrocytes was determined using the CCK-8 assay. Data represent the mean±SEM of three independent experiments.

In the $H_2O_2$ group, mRNA levels of inflammatory markers (IL-6, COX-2, iNOS) and catabolic factors (MMP3, MMP13, ADAMTS5) were significantly upregulated compared to the NC group, while anabolic markers (aggrecan, collagen-II) were downregulated (Fig. 3a–c, $p < 0.05$). Notably, hUCMSCs-dECM treatment ($H_2O_2$+dECM) more effectively suppressed these senescence-associated changes than commercial ECM ($H_2O_2$+commercial ECM), particularly in reducing IL-6 and MMP13 expression (Fig. 3b, c). Immunofluorescence confirmed that hUCMSCs-dECM significantly reversed $H_2O_2$-induced overexpression of IL-1β, IL-6, MMP3, and MMP13 (Fig. 3d–f, $p < 0.05$). Western blot analysis further validated these findings: hUCMSCs-dECM normalized protein levels of IL-6, COX-2, iNOS, MMP3, MMP13, and ADAMTS5 elevated by $H_2O_2$, with stronger inhibition of MMP13 and ADAMTS5 compared to commercial ECM (Fig. 3g–i, $p < 0.05$).

Critically, hUCMSCs-dECM alone (dECM group) exhibited superior antioxidant activity to commercial ECM, as evidenced by lower baseline expression of senescence markers relative to NC (Fig. 3a–c). These results collectively demonstrate that hUCMSCs-dECM not only mitigates oxidative stress-induced chondrocyte senescence but also surpasses commercial ECM in preserving matrix homeostasis and suppressing inflammatory/catabolic responses.

### 3.4. hUCMSCs-dECM attenuates $H_2O_2$-induced senescence and inhibits chondrocyte SASP depending on STING

Previous studies have demonstrated markedly increased STING levels in both human and murine OA tissues as well as IL-1β-stimulated chondrocytes. Elevated STING activity not only upregulates MMP13 and ADAMTS5 but also inhibits the expression of Aggrecan and Collagen II production while exacerbating cellular apoptosis and senescence processes [32]. To investigate dECM's potential anti-inflammatory mechanism through STING regulation, siRNA-mediated STING silencing was implemented. Experimental data from RT-qPCR and Western blot analyses revealed that dECM administration effectively reduced pro-inflammatory markers (IL-6, COX-2, iNOS) and cartilage-degrading enzymes (MMP3, MMP13, ADAMTS5) while enhancing matrix components (Aggrecan, Collagen II) compared to the $H_2O_2$ group. Notably, these beneficial effects were completely negated following STING knockdown. Immunofluorescence results also showed that dECM decreased the expression of IL-1β, IL-6, MMP3, and MMP13 in oxidatively stressed chondrocytes, with these protective effects being similarly abolished by STING silencing (Fig 4d-f, $p < 0.05$). Collectively, these findings establish that the therapeutic ability of dECM to inhibit chondrocyte SASP phenotypes and promote extracellular matrix synthesis depends on STING.

### 3.5. hUCMSCs-dECM inhibits NF-κB pathway activation in senescent chondrocytes

Prior studies link STING to NF-κB activation, with NF-κB inhibition mitigating STING-driven apoptosis and ECM metabolic dysregulation [43,44]. Therefore, we investigated the effect of hUCMSCs-dECM on the NF-κB pathway in senescent chondrocytes. Protein blotting results showed that dECM inhibited IL-1β-induced phosphorylation of IKKβ, IκBα, and p65 as well as IκBα degradation compared to controls, whereas siRNA-STING eliminated the inhibitory effect of dECM on the NF-κB pathway (Fig 5a-e). These results indicated that hUCMSCs-dECM significantly inhibited the activation of the NF-κB pathway in senescent chondrocytes.

### 3.6. hUCMSCs-dECM inhibits chondrocyte senescence in rats *in vivo*

Previous studies have identified abundant senescent cells in rat cartilage after ACLT. To evaluate the *in vivo* efficacy of dECM on senescent chondrocytes, we generated a rat OA model by ACLT surgery. Immunohistochemical and quantitative analyses showed that intra-articular injection of dECM inhibited the expression of senescence markers P16 and P21, alongside inflammatory mediators iNOS, and COX2, (Fig 6a-d, $p < 0.05$). Similarly, dECM treatment markedly reduced the levels of SASP biomarkers MMP3 and IL-6 in the ACLT group (Fig 7a-d, $p < 0.05$). These results collectively demonstrate that dECM attenuates chondrocyte senescence and SASP *in vivo*, underscoring its therapeutic potential for OA management.

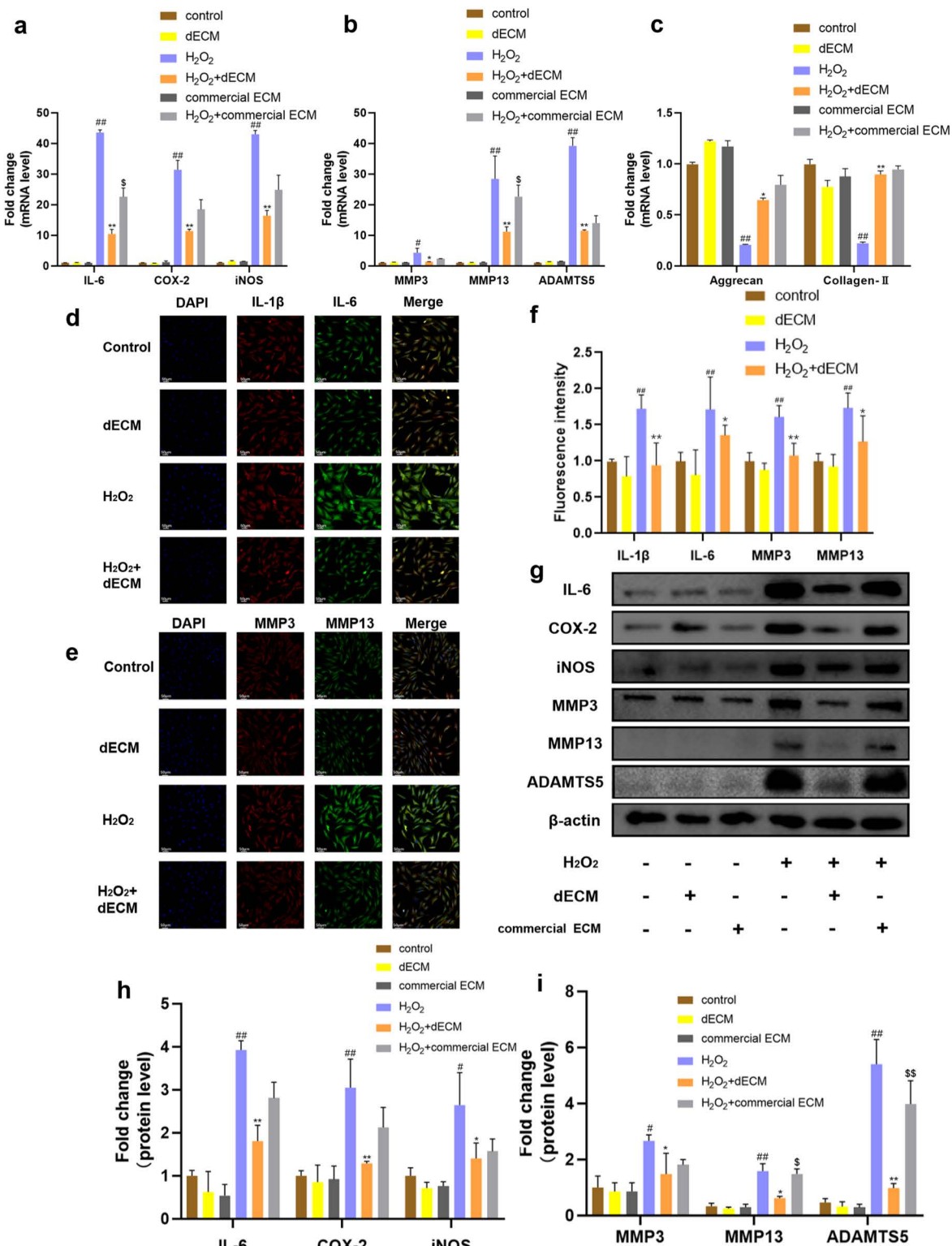

**Fig 3. Effects of hUCMSCs-dECM on chondrocytes caused by H₂O₂ intervention. (a-c)** RT-qPCR analysis of IL-6, COX-2, iNOS, MMP3, MMP13, ADAMTS5, aggrecan and collagen-II in chondrocytes (n = 3); **(d-f)** Immunofluorescence and semi-quantitative analysis of MMP3, MMP13, IL-1β,IL-6 (n = 5, 200×); **(g-i)** Western blot and semi-quantitative analysis of IL-6, COX-2, iNOS, MMP3, MMP13, and ADAMTS5 (n = 3). ## $p < 0.01$ vs. NC; # $p < 0.05$ vs. NC; ** $p < 0.01$ vs. H₂O₂ group; * $p < 0.05$ vs. H₂O₂ group; $ $p < 0.05$ vs. H₂O₂ + dECM group; $$ $p < 0.01$ vs. H₂O₂ + dECM group.

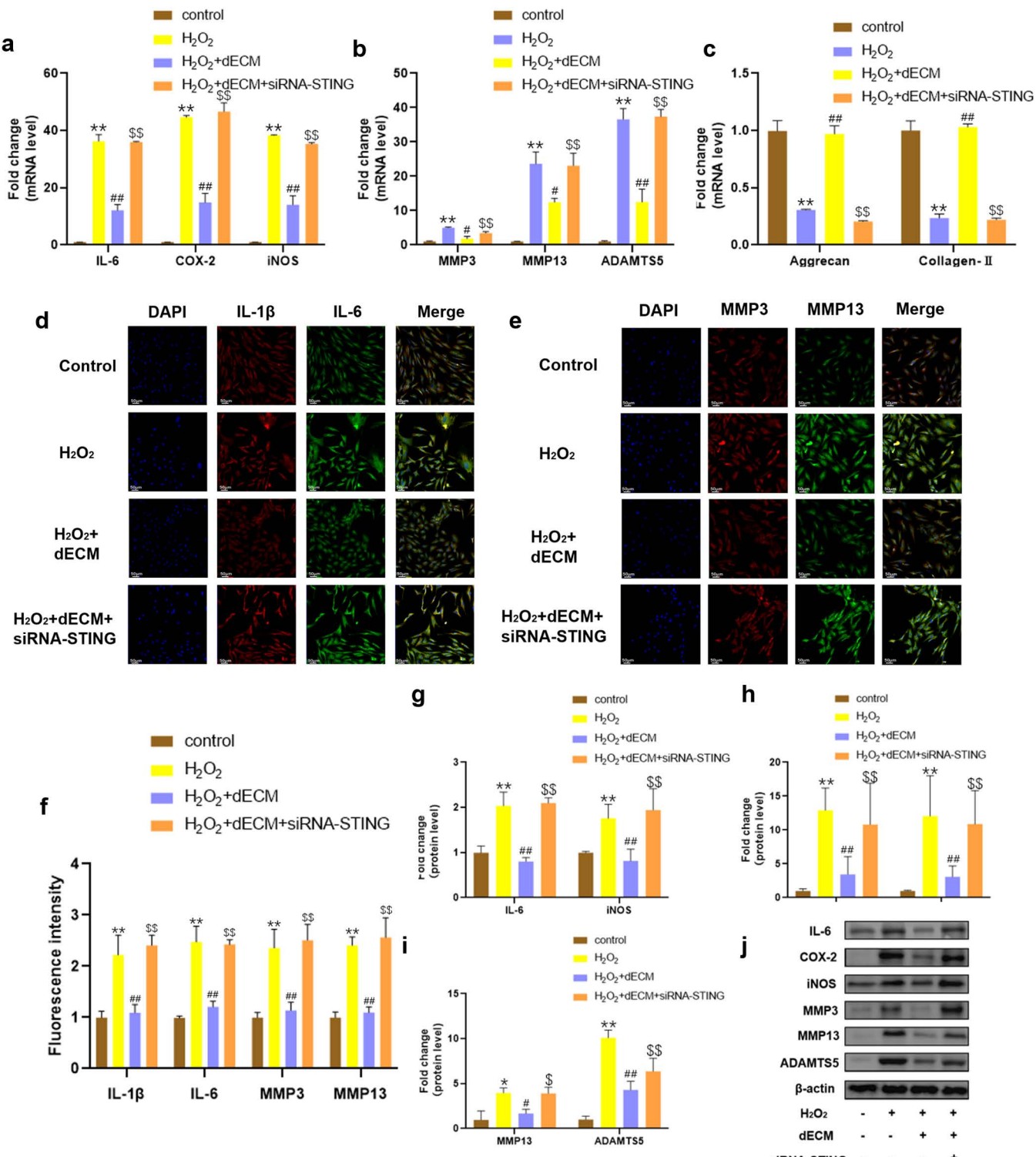

**Fig 4. Effects of hUCMSCs-dECM on inflammation in STING knocked down chondrocytes. (a-c)** RT-qPCR analysis of IL-6, COX-2, iNOS, MMP3, MMP13, ADAMTS5, aggrecan and collagen-II in chondrocytes (n=3); **(d-g)** Immunofluorescence and semi-quantitative analysis of MMP3, MMP13, IL-1β,IL-6 (n=5, 200×); (g-i) Western blot and semi-quantitative analysis of IL-6, COX-2, iNOS, MMP3, MMP13, ADAMTS5 (n=3). $*$ $p < 0.05$ vs. NC; $**$ $p < 0.01$ vs. NC; $\#$ $p < 0.05$ vs. $H_2O_2$ group; $\#\#$ $p < 0.01$ vs. $H_2O_2$ group; $\$$ $p < 0.05$ vs. $H_2O_2 + dECM$ group; $\$\$$ $p < 0.01$ vs. $H_2O_2 + dECM$ group.

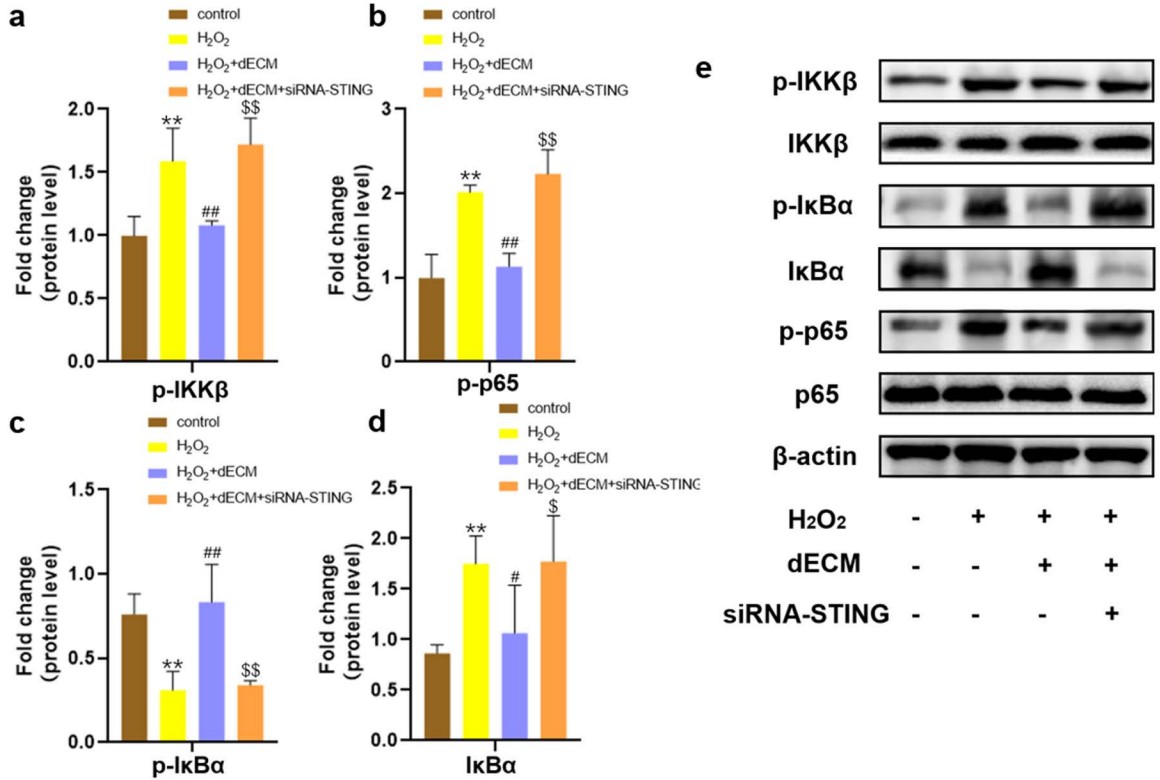

**Fig 5. Effects of hUCMSCs-dECM on the NF-κB pathway in STING knocked down chondrocytes. (a-e)** Western blot and semi-quantitative analysis of p-IKKβ, IKKβ, p-IκBα, IκBα, p-p65, and p65 (n = 3). * $p < 0.05$ vs. NC; ** $p < 0.01$ vs. NC; # $p < 0.05$ vs. H₂O₂ group; ## $p < 0.01$ vs. H₂O₂ group; $ $p < 0.05$ vs. H₂O₂ + dECM group; $$ $p < 0.01$ vs. H₂O₂ + dECM group.

### 3.7. hUCMSCs-dECM can protect cartilage in vivo

To evaluate the NF-κB signaling pathway activation in rat chondrocytes under *in vivo* conditions, protein expression of total p65 and its phosphorylated form (p-p65) was quantitatively analyzed. Immunohistochemical results showed that dECM treatment significantly attenuated p-p65 protein expression levels (Fig 7c, $p < 0.05$), though total p65 expression remained consistent across experimental groups (Fig 7d, $p > 0.05$). Hematoxylin-eosin and safranin O-fast green staining results showed severe cartilage surface erosion and proteoglycan depletion were observed in the ACLT group, whereas hUCMSCs-dECM intervention substantially mitigated these degenerative changes, evidenced by preserved cartilage architecture and significantly reduced OARSI scores (Fig 7e, $p < 0.01$). These collective findings confirm that hUCMSCs-dECM exerts cartilage-protective effects *in vivo* through modulation of NF-κB pathway activity.

## 4. Discussion

Osteoarthritis, as a common and disabling disease, the first line of treatment remains with non-pharmacological treatments such as health education and self-management [45]. In OA, the structural destruction of articular cartilage is particularly significant. Repair or regeneration of articular cartilage has become a hot topic in the treatment of osteoarthritis [46–48]. Evidence from both in vitro and in vivo has shown that eliminating or delaying chondrocyte senescence is effective in slowing down the pathogenesis and progression of OA, and in helping patients with OA [8,49,50]. As a new therapy, hUCMSCs implantation has shown promising therapeutic effects on OA in both animal and human studies, and the

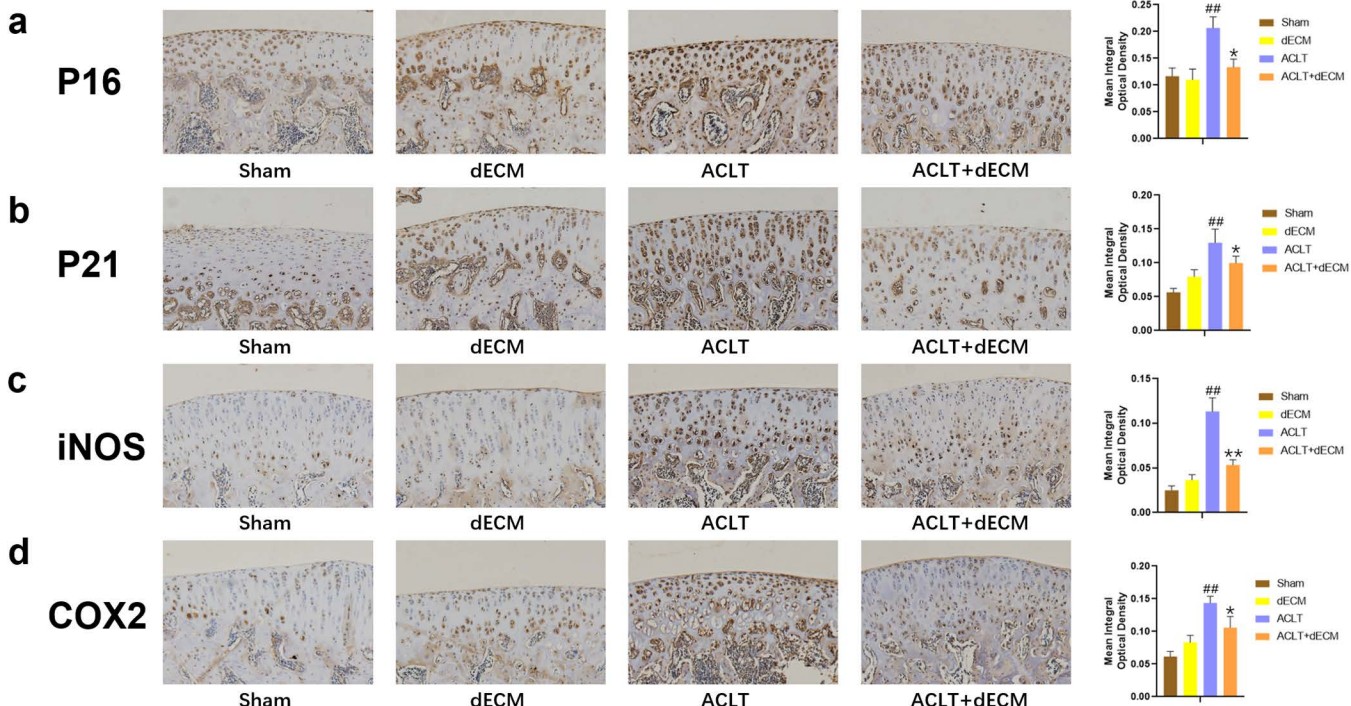

**Fig 6. hUCMSCs-dECM attenuated senescence of chondrocytes in vivo.** (a–d) Immunohistochemical staining and quantitative results of P16, P21, iNOS, and COX2 in the synovium (n = 5). ## $p < 0.01$ vs. sham; * $p < 0.05$ vs. ACLT; ** $p < 0.01$ vs. ACLT.

secretion products of hUCMSCs promote endogenous repair after intra-articular injection of the hUCMSCs [14,15,51,52]. Therefore, we wished to demonstrate that hUCMSC secretory products have a therapeutic effect on senescent chondrocytes in OA.Our study demonstrates that hUCMSCs-dECM effectively alleviates chondrocyte senescence and suppresses SASP by targeting the STING-NF-κB pathway. This finding builds upon prior reports that dECM-based therapies mitigate tissue damage [42,53], but distinctively, we establish hUCMSCs-dECM as a novel anti-senescence agent with superior antioxidant and matrix-preserving effects compared to commercial ECM (Fig. 3). Specifically, hUCMSCs-dECM reduced $H_2O_2$-induced inflammatory (IL-6, COX-2, iNOS) and catabolic (MMP3, MMP13, ADAMTS5) markers, outperforming commercial ECM in key metrics (Fig. 3b–i). These results align with emerging evidence that decellularized matrices retain bioactive cues to counteract oxidative stress [53], yet our work uniquely links these effects to STING-NF-κB modulation.

Mechanistically, we identified the STING-NF-κB axis as central to hUCMSCs-dECM's therapeutic action. While Guo et al. [27] implicated STING in chondrocyte senescence, our study advances this paradigm by showing that hUCMSCs-dECM directly suppresses STING activation, thereby inhibiting NF-κB-driven SASP (Fig. 4). This contrasts with prior dECM strategies focused solely on structural repair [53], highlighting our dual mechanism: senolytic activity (reducing p16/p21) and matrix restoration (enhancing collagen-II/aggrecan).

hUCMSCs are multipotent cells derived from umbilical cord tissue. Owing to their high proliferative capacity, ease of accessibility, and low immunogenicity, hUCMSCs have emerged as a prominent candidate for cell-based therapies in OA treatment [54]. Despite their therapeutic potential, clinical translation of hUCMSCs faces challenges including efficient large-scale production, long-term preservation, and risks of unintended differentiation within joint microenvironments [47,55]. Recent advances highlight the utility of decellularized matrices in restoring tissue structure and function [53]. For instance, Yin et al. demonstrated that chondrocyte-derived decellular matrix effectively regenerate damaged cartilage

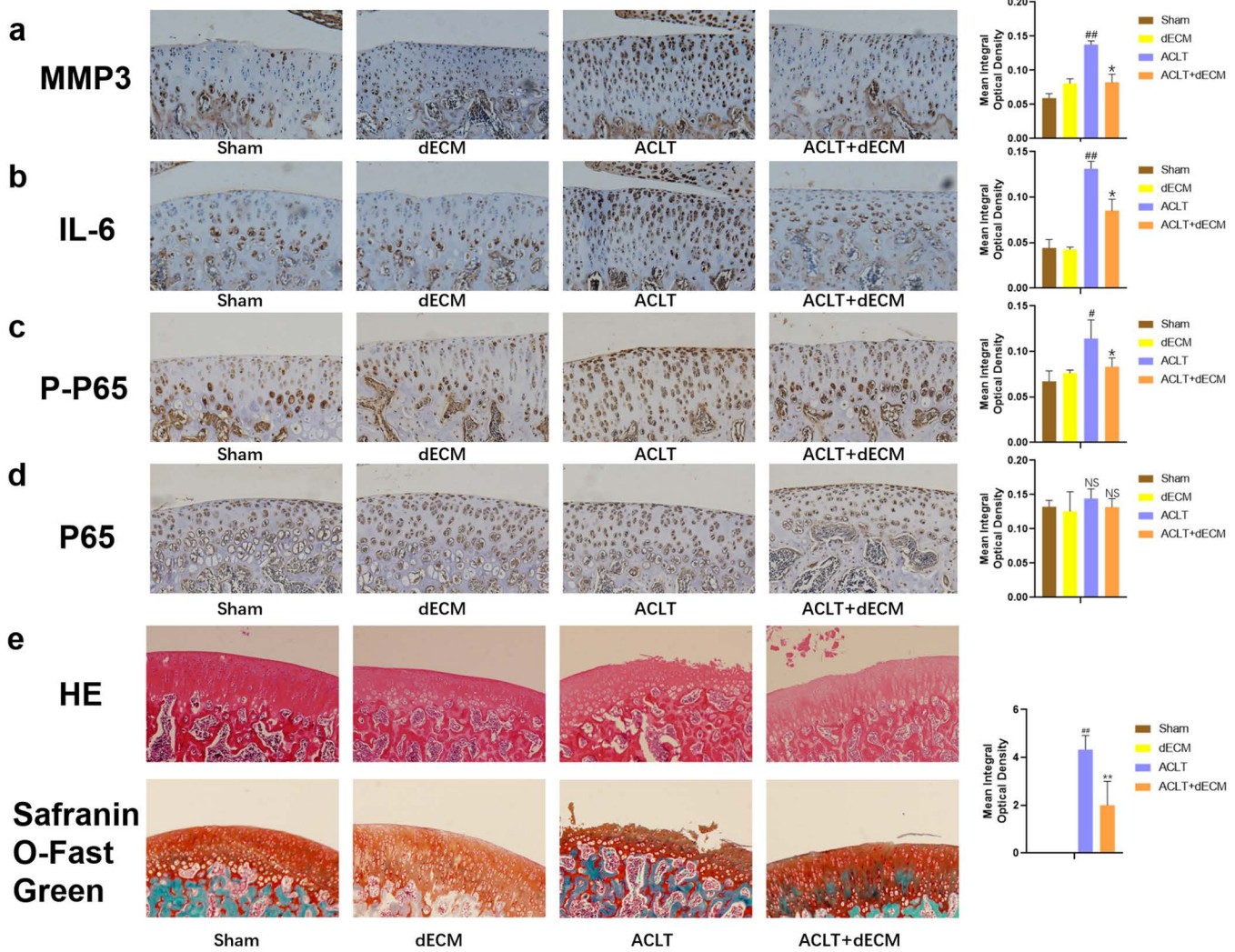

**Fig 7. hUCMSCs-dECM inhibits SASP, inactivates the NF-κB signaling pathway, and protects cartilage in vivo.** (a-b) Immunohistochemical staining and quantitative results of MMP3 and IL-6 (n = 5). (c-d) Immunohistochemical staining and quantitative results of p65 and p-p65 (n = 5). (e) Representative pictures of HE, safranin O-fast green staining, and OARSI score for each group (n = 5). # $p < 0.05$ vs. sham; ## $p < 0.01$ vs. sham; * $p < 0.05$ vs. ACLT; ** $p < 0.01$ vs. ACLT.

[56]. Building on this, we extracted the decellular matrix from hUCMSCs to investigate its therapeutic effects on senescent chondrocytes both *in vivo* and *in vitro*. Our results demonstrated that hUCMSCs-dECM attenuated chondrocyte oxidative stress-associated senescence and enhanced matrix synthesis *in vitro* and *in vivo*. Notably, hUCMSCs-dECM exhibited no cytotoxicity even at high concentrations (200 µg/mL), addressing concerns about stem cell-derived products inducing unintended differentiation [43,51]. Unlike cell-based therapies requiring viability control [14,49], our acellular approach simplifies clinical translation while retaining therapeutic potency.

Activation of the cGAS-STING pathway has been shown to induce senescence. Dou et al. reported that STING knockdown suppresses senescence [57], while Guo et al. observed elevated STING levels in senescent chondrocytes and demonstrated that STING silencing reduces senescence via NF-κB signaling modulation [32]. Specifically, STING promotes chondrocyte senescence by activating NF-κB signaling, as evidenced by cGAS-STING pathway regulation. Our

findings establish that hUCMSCs-dECM inhibits chondrocyte senescence by targeting STING-dependent NF-κB signaling pathway suppression, offering a mechanistically grounded strategy for OA intervention.

There are still some limitations in this study. OA as a whole joint disease, the therapeutic effect needs to be fully demonstrated by studying the effect of hUCMSCs-dECM on other tissues in the joint, such as synovial tissues. Besides, while hUCMSCs-dECM attenuated chondrocyte senescence, OA involves crosstalk between cartilage, synovium, and subchondral bone [41]. Future studies should evaluate dECM's effects on synovial inflammation and bone remodeling. Although we linked STING-NF-κB inhibition to senescence alleviation, upstream regulators and downstream effectors warrant deeper exploration.

## 5. Conclusion

In summary, hUCMSCs-dECM represents a promising acellular therapy for OA, combining antioxidant, anti-senescent, and matrix-regenerative properties. By targeting the STING-NF-κB pathway, it addresses both cellular dysfunction and structural degeneration—a dual advantage over conventional ECM products. Further validation in large-animal OA models and mechanistic studies will accelerate its translational potential.

## Supporting information

**S1 Table. Primer sequences used in this study.**
(DOCX)

## Acknowledgments

Thanks to Lina Zhou, Yingxia Jin, Lili Li and Yuan He (Central Laboratory, Renmin Hospital of Wuhan University) for their assistance in this study.

## Author contributions

**Conceptualization:** Yuelong Zhang, Xunshan Ren, Huangming Zhuang, Junming Zhu, Panghu Zhou.

**Formal analysis:** Miradj Siddick Adam.

**Funding acquisition:** Panghu Zhou.

**Investigation:** Yuelong Zhang, Xunshan Ren, Huangming Zhuang, Huajie Li, Rongling Feng.

**Methodology:** Yuelong Zhang, Miradj Siddick Adam.

**Supervision:** Junming Zhu, Panghu Zhou.

**Writing – original draft:** Panghu Zhou.

**Writing – review & editing:** Yuelong Zhang.

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
