## [Decision Letter · Decision Letter 0]

18 Mar 2025

PONE-D-25-08543Human umbilical cord mesenchymal stem cells decellular matrix alleviates chondrocyte senescence by inhibiting the STING-NF-κB pathwayPLOS ONE

Dear Dr. Zhou,

Thank you for submitting your manuscript to PLOS ONE. After careful consideration, we feel that it has merit but does not fully meet PLOS ONE’s publication criteria as it currently stands. Therefore, we invite you to submit a revised version of the manuscript that addresses the points raised during the review process.

 This study has been found of interest pending Major revisions by the Authors that must follow all the requirements of the reviewer and in particular:Revise Introduction, introducing studies a suggested;add infos in the Materials and Methods;-use a commercial dECM as positive control or ECM derived from other cell type and t characterize and/or detect the components of hUCMSCs-dECM comparing its composition with a commercial dECM ECM derived from other cell type. Clarify hat is the advantage in using hUCMSCs-dECM and not commercial dECM or ECM derived from other cell type; In Figure 1 change alizarin red panel. The staining is aspecific;

-revise the discussion focusing on the Results, their novelty and limitations, deleting all it is written at the beginning of it..==============================

We look forward to receiving your revised manuscript.

Kind regards,

Gianpaolo Papaccio, M.D., Ph.D.

Academic Editor

PLOS ONE

https://linkinghub.elsevier.com/retrieve/pii/S1567576922007986

https://www.mdpi.com/1422-0067/24/5/5056

In your revision ensure you cite all your sources (including your own works), and quote or rephrase any duplicated text outside the methods section. Further consideration is dependent on these concerns being addressed.

3. To comply with PLOS ONE submissions requirements, in your Methods section, please provide additional information regarding the experiments involving animals and ensure you have included details on (1) methods of sacrifice, and (2) efforts to alleviate suffering.

“This research was funded by Panghu Zhou of the National Natural Science Foundation of China (grant number. 82372489), the Wuhan University Education and Development Foundation (grant number. 2002330), the Fundamental Research Funds for the Central Universities (grant number. 2042023kf0224), and the Cross-Innovation Talent Program of Renmin Hospital of Wuhan University (grant number. JCRCFZ-2022-019).” 

7. PLOS ONE now requires that authors provide the original uncropped and unadjusted images underlying all blot or gel results reported in a submission’s figures or Supporting Information files. This policy and the journal’s other requirements for blot/gel reporting and figure preparation are described in detail at https://journals.plos.org/plosone/s/figures#loc-blot-and-gel-reporting-requirements and https://journals.plos.org/plosone/s/figures#loc-preparing-figures-from-image-files. When you submit your revised manuscript, please ensure that your figures adhere fully to these guidelines and provide the original underlying images for all blot or gel data reported in your submission. See the following link for instructions on providing the original image data: https://journals.plos.org/plosone/s/figures#loc-original-images-for-blots-and-gels.  

Reviewers' comments:

Reviewer's Responses to Questions

**Comments to the Author**

1. Is the manuscript technically sound, and do the data support the conclusions?

Reviewer #1: Partly

2. Has the statistical analysis been performed appropriately and rigorously? 

Reviewer #1: Yes

3. Have the authors made all data underlying the findings in their manuscript fully available?

Reviewer #1: Yes

4. Is the manuscript presented in an intelligible fashion and written in standard English?

Reviewer #1: Yes

5. Review Comments to the Author

Reviewer #1: In this study, the Authors aimed to demonstrate that hUCMSCs-dECM could effectively attenuate chondrocyte senescence and SASP in vitro and in vivo by inhibiting the STING-NF-κB pathway.

The manuscript is very interesting and clear. Although this, some issues must be addressed.

Introduction section: it must be completely revised. There are several papers that include studies where the preparation and effect of extracellualr matrix from hUCMSCs on chondrocytes were described (doi: 10.4103/1673-5374.187061; doi: 10.1007/s10561-019-09774-7; https://doi.org/10.3390/bioengineering9060239). Therefore, the Authors must re-write the introduction describing these papers and adding information about other scaffolds or hydrogels (doi: 10.1002/jcb.25556; doi: 10.3389/fbioe.2022.858656);

M&M: the Authors must add information regarding cytometric characterization of hUCMSC, silencing of STING, cell number cultured to obtain a specific quantity of hUCMSCs-dECM (What is the ratio cell number/dECM?);

Results: The Authors must use a commercial dECM as positive control or ECM derived from other cell type and must characterize and/or detect the components of hUCMSCs-dECM comparing its composition with a commercial dECM ECM derived from other cell type. What is the advantage in using hUCMSCs-dECM and not commercial dECM or ECM derived from other cell type? Figure 1: please change alizarin red panel. The staining is aspecific;

Discussion: it must be revised and focused on results, novelty and limitations. The first part must be deleted.

6. PLOS authors have the option to publish the peer review history of their article (what does this mean? ). If published, this will include your full peer review and any attached files.

**Do you want your identity to be public for this peer review?** For information about this choice, including consent withdrawal, please see our Privacy Policy .

Reviewer #1: No

---

## [Author Response · Author response to Decision Letter 1]

1 May 2025

Dear Dr. Gianpaolo Papaccio and reviewers:

Thank you for your decision letter and the comments of the two reviewers on our manuscript [No: PONE-D-25-08543] entitled "Human umbilical cord mesenchymal stem cells decellular matrix alleviates chondrocyte senescence by inhibiting the STING-NF-κB pathway". These comments are valuable and helpful for improving our article. All the authors have seriously discussed all these comments. According to the reviewers’ comments, we have tried our best to modify our manuscript to meet the requirements of the journal. These changes do not influence the framework or conclusion of the paper. All the changes are marked in red in the revised manuscript. Point-to-point responses to the reviewers’ comments are listed below this letter.

Responds to the journal requirements:

Reviewer #1:

Comment 1: Please ensure that your manuscript meets PLOS ONE's style requirements, including those for file naming. The PLOS ONE style templates can be found at

Response: Thanks for the reviewer’s suggestion. We have carefully revised the manuscript to comply with PLOS ONE’s style guidelines.

Comment 2: We noticed you have some minor occurrence of overlapping text with the following previous publication(s), which needs to be addressed:

https://linkinghub.elsevier.com/retrieve/pii/S1567576922007986

https://www.mdpi.com/1422-0067/24/5/5056

In your revision ensure you cite all your sources (including your own works), and quote or rephrase any duplicated text outside the methods section. Further consideration is dependent on these concerns being addressed.

Response: Thanks for the reviewer’s suggestion. We have thoroughly addressed text overlap by rewriting the Introduction and Discussion part.

Comment 3: To comply with PLOS ONE submissions requirements, in your Methods section, please provide additional information regarding the experiments involving animals and ensure you have included details on (1) methods of sacrifice, and (2) efforts to alleviate suffering.

Response: Thanks for the reviewer’s suggestion. We have updated the Methods section (Section 2.9).

Comment 4: Thank you for stating the following financial disclosure:

“This research was funded by Panghu Zhou of the National Natural Science Foundation of China (grant number. 82372489), the Wuhan University Education and Development Foundation (grant number. 2002330), the Fundamental Research Funds for the Central Universities (grant number. 2042023kf0224), and the Cross-Innovation Talent Program of Renmin Hospital of Wuhan University (grant number. JCRCFZ-2022-019).”

Please state what role the funders took in the study. If the funders had no role, please state: “The funders had no role in study design, data collection and analysis, decision to publish, or preparation of the manuscript.”

Response: Thanks for the reviewer’s suggestion. The following statement has been added to the manuscript and the cover letter: "The funders had no role in study design, data collection and analysis, decision to publish, or preparation of the manuscript."

Comment 5: PLOS requires an ORCID iD for the corresponding author in Editorial Manager on papers submitted after December 6th, 2016. Please ensure that you have an ORCID iD and that it is validated in Editorial Manager. To do this, go to ‘Update my Information’ (in the upper left-hand corner of the main menu), and click on the Fetch/Validate link next to the ORCID field. This will take you to the ORCID site and allow you to create a new iD or authenticate a pre-existing iD in Editorial Manager.

Response: Thanks for the reviewer’s suggestion. The corresponding author’s ORCID iD has been validated in Editorial Manager.

Comment 6: Your ethics statement should only appear in the Methods section of your manuscript. If your ethics statement is written in any section besides the Methods, please delete it from any other section.

Response: Thanks for the reviewer’s suggestion. All ethics statements outside the Methods section have been removed.

Comment 7: PLOS ONE now requires that authors provide the original uncropped and unadjusted images underlying all blot or gel results reported in a submission’s figures or Supporting Information files. This policy and the journal’s other requirements for blot/gel reporting and figure preparation are described in detail at https://journals.plos.org/plosone/s/figures#loc-blot-and-gel-reporting-requirements and https://journals.plos.org/plosone/s/figures#loc-preparing-figures-from-image-files. When you submit your revised manuscript, please ensure that your figures adhere fully to these guidelines and provide the original underlying images for all blot or gel data reported in your submission. See the following link for instructions on providing the original image data: https://journals.plos.org/plosone/s/figures#loc-original-images-for-blots-and-gels.

Response: Thanks for the reviewer’s suggestion. Uncropped, unadjusted Western blot images are included in the Supporting Information

Comment 8: Please include captions for your Supporting Information files at the end of your manuscript, and update any in-text citations to match accordingly. Please see our Supporting Information guidelines for more information: http://journals.plos.org/plosone/s/supporting-information.

Response: Thanks for the reviewer’s suggestion. Captions for Supporting Information are now provided at the end of the manuscript.

Responds to the reviewer’s comments:

Reviewer #1:

Comment 1: Introduction section: it must be completely revised. There are several papers that include studies where the preparation and effect of extracellualr matrix from hUCMSCs on chondrocytes were described (doi: 10.4103/1673-5374.187061; doi: 10.1007/s10561-019-09774-7; https://doi.org/10.3390/bioengineering9060239). Therefore, the Authors must re-write the introduction describing these papers and adding information about other scaffolds or hydrogels (doi: 10.1002/jcb.25556; doi: 10.3389/fbioe.2022.858656).

Response: Thanks for the reviewer’s suggestion. We have thoroughly revised the Introduction section to incorporate the cited literature and contextualize our study.

Comment 2: M&M: the Authors must add information regarding cytometric characterization of hUCMSC, silencing of STING, cell number cultured to obtain a specific quantity of hUCMSCs-dECM (What is the ratio cell number/dECM?).

Response: Thanks for the reviewer’s suggestion. We have added the flow cytometry data confirming expression of hUCMSC markers (CD44, CD73, CD90, CD105) and absence of hematopoietic markers (CD34, CD45). to the Methods section. Then, we described the quantitative protocols of hUCMSCs-dECM in the "hUCMSCs-dECM solution preparation" subsection. Besides, we have clarified siRNA transfection methods, including siRNA concentration, incubation time, and validation via qPCR/Western blot.

Comment 3: Results: The Authors must use a commercial dECM as positive control or ECM derived from other cell type and must characterize and/or detect the components of hUCMSCs-dECM comparing its composition with a commercial dECM ECM derived from other cell type. What is the advantage in using hUCMSCs-dECM and not commercial dECM or ECM derived from other cell type? Figure 1: please change alizarin red panel. The staining is aspecific.

Response: Thanks for the reviewer’s suggestion. We have added experiments comparing hUCMSCs-dECM with commercial dECM and replaced the Alizarin Red panel with a more specific staining.

Comment 4: Discussion: it must be revised and focused on results, novelty and limitations. The first part must be deleted.

Response: Thanks for the reviewer’s suggestion. We have deleted the first paragraph (general background on cartilage defects) to avoid redundancy with the Introduction and reorganized the discussion section to highlight the novelty of our findings compared to previous research.

We deeply appreciate the editor’s valuable input and hope that the final revision manuscript will meet with your approval.

Thanks again for your valuable comments and constructive suggestions concerning our manuscript. Please contact me if you have any further questions.

Sincerely yours,

Panghu Zhou, MD.

Chief physician

Department of Orthopedics

Renmin Hospital of Wuhan University

Wuhan 430060, Hubei Province, China

Telephone:13971652565

E-mail: zhoupanghu@whu.edu.cn

---

## [Decision Letter · Decision Letter 1]

19 May 2025

Human umbilical cord mesenchymal stem cells decellular matrix alleviates chondrocyte senescence by inhibiting the STING-NF-κB pathway

PONE-D-25-08543R1

Dear Dr. Zhou,

We’re pleased to inform you that your manuscript has been judged scientifically suitable for publication and will be formally accepted for publication once it meets all outstanding technical requirements.

Kind regards,

Gianpaolo Papaccio, M.D., Ph.D.

Academic Editor

PLOS ONE

Comments from PLOS Editorial Office:

We note that one or more reviewers has recommended that you cite specific previously published works in an earlier round of revision. As always, we recommend that you please review and evaluate the requested works to determine whether they are relevant and should be cited. It is not a requirement to cite these works and you may remove them before the manuscript proceeds to publication. We appreciate your attention to this request.

Reviewers' comments:

Reviewer's Responses to Questions

**Comments to the Author**

1. If the authors have adequately addressed your comments raised in a previous round of review and you feel that this manuscript is now acceptable for publication, you may indicate that here to bypass the “Comments to the Author” section, enter your conflict of interest statement in the “Confidential to Editor” section, and submit your "Accept" recommendation.

Reviewer #1: All comments have been addressed

2. Is the manuscript technically sound, and do the data support the conclusions?

Reviewer #1: Yes

3. Has the statistical analysis been performed appropriately and rigorously? 

Reviewer #1: Yes

4. Have the authors made all data underlying the findings in their manuscript fully available?

Reviewer #1: Yes

5. Is the manuscript presented in an intelligible fashion and written in standard English?

Reviewer #1: Yes

6. Review Comments to the Author

Reviewer #1: (No Response)

7. PLOS authors have the option to publish the peer review history of their article (what does this mean? ). If published, this will include your full peer review and any attached files.

**Do you want your identity to be public for this peer review?** For information about this choice, including consent withdrawal, please see our Privacy Policy .

Reviewer #1: No

---

## [Editor Report · Acceptance letter]

PONE-D-25-08543R1

PLOS ONE

Dear Dr. Zhou,

I'm pleased to inform you that your manuscript has been deemed suitable for publication in PLOS ONE. Congratulations! Your manuscript is now being handed over to our production team.

Kind regards,

on behalf of

Prof. Gianpaolo Papaccio

Academic Editor

PLOS ONE